# Dispositional Awe and Self-Worth in Chinese Undergraduates: The Suppressing Effects of Self-Concept Clarity and Small Self

**DOI:** 10.3390/ijerph20136296

**Published:** 2023-07-04

**Authors:** Benyu Zhang, Rongmao Lin

**Affiliations:** School of Psychology, Fujian Normal University, Fuzhou 350117, China; zhangbenyu@fjnu.edu.cn

**Keywords:** awe, self-concept, self-esteem, small self, self-transcendence

## Abstract

Background: The positive role of dispositional awe has been seen in personality and in health. However, its impact on self-worth and internal mechanisms have been unclear. Purposes: This study explored the relationship between dispositional awe and self-worth and the roles of self-concept clarity and the small self in this association. Methods: With a cluster sampling, a cross-sectional sample of 1888 Chinese undergraduates were recruited from Fuzhou, a southeast coastal city in the P.R.C. All the data were analyzed with Pearson’s correlations and the structural equation model (SEM) based on SPSS 25.0 and Mplus 8.1. Results: Dispositional awe was positively correlated with both personal-oriented and social-oriented self-worth (*r_s_* = 0.12, 0.27) and was also positively correlated with small self (*r* = 0.33) but negatively correlated with self-concept clarity (*r* = −0.18); in the full model, the direct effect of dispositional awe on society-oriented self-worth was 0.36 (75%); the indirect effects of small self and self-concept clarity were −0.09 (18.8%) and −0.01 (2.1%), respectively; and the chain indirect effect was −0.02 (4.2%). Similarly, the direct effect of dispositional awe on person-oriented self-worth was 0.50 (83.3%); the indirect effects of small self and self-concept clarity were −0.07 (11.7%) and −0.01 (1.7%), respectively; and the chain indirect effect was −0.02 (3.3%); all the indirect effects were suppressing effects, for they were contrary to the direct effects. Conclusion: This study suggested that dispositional awe could help people better understand themselves and enhance their sense of self-worth.

## 1. Introduction

As an important personality trait, dispositional awe describes a social reaction tendency for people to respect and fear external things because of their size, authority, sacredness, or mystique, beyond the scope of individual ability or understanding [1,2]. Studies have shown that dispositional awe is helpful in stimulating individual creativity [3], changing individual time perception [4], improving individual subjective and career well-being [5], and increasing individual collective investment and prosocial behavior [6]. Individuals with higher dispositional awe can better understand themselves and the limitations of existence, maintain consistency with the collective and society, and engage more actively in collective or social activities. Although the positive effects of dispositional awe have drawn a lot of attention, its role in self-worth, the core of the self-concept system, and its mechanisms remain unknown. Through a cross-sectional study, this study intends to investigate the correlation between dispositional awe and self-worth, as well as the role of self-concept clarity and sense of small self between them, which further test the role of dispositional awe in undergraduates’ self-awareness and emotion and provide some suggestions for undergraduates’ mental health.

### 1.1. Whether Dispositional Awe Is Associated with Self-Worth Positively or Negatively

Self-worth refers to a subjective judgment and experience in certain situations [7]. It is the core of the self-concept system and is frequently explained in conjunction with self-esteem [8,9]). As a set of characteristics based on an individual’s ability to evaluate themselves and their attitudes towards “self” formed on this basis, self-worth and its contingency can facilitate self-regulation [10], enhance self-esteem [11,12,13], and further reduce self-criticism [14], depressive symptoms [15], and self-injury [14]. Although self-worth is usually regarded as a stable trait, it is also easily affected by context and time when individuals evaluate their own abilities and characteristics [16,17]. In different social situations, people may have different judgments and experiences with their own self-values. Huang et al. pointed out that the general self-worth of Chinese people can be divided into two aspects: personal-oriented and social-oriented self-worth [18,19]. The former refers to a sense of self-worth that is based on the development and improvement of personal potential by oneself, while the latter is a sense of self-worth that depends on external approval and others’ appraisals [18,19].

It is unclear whether dispositional awe is positively or negatively correlated with self-worth. The superficial explanation is the small-self hypothesis of dispositional awe [20,21]. It supports the idea that dispositional awe, stemming from the vastness of external things [1], is associated with a sense that one is a part of something larger than oneself, triggering an almost metaphorical sense of the smallness of the self [20,21]. The smaller self may have a negative correlation with higher self-worth.

The smaller and diminished self is a transitory state when people are in awesome situations; however, the self-transcendent quality is the essence of dispositional awe [22,23]. The positive effect of dispositional awe may be shown in the ability of individuals to transcend themselves [24]. Specifically, greater dispositional awe can urge people to overcome their existing feeling of smallness [25], seek to connect with the others and world [26], focus more on collective input and the authentic self [27], and thus achieve and be aware of greater self-worth. The self-transcendent character of dispositional awe is also demonstrated in that it can release daily stress and increase life satisfaction [5].

The primary purpose of our study was to test whether dispositional awe is positively or negatively associated with self-worth. Based on the self-transcendent theory, we tend to hypothesize a positive association between them and hold up Hypothesis 1:

**Hypothesis** **1:**
*Greater dispositional awe is positively associated with greater self-worth.*


### 1.2. The Indirect Roles of Self-Concept Clarity and Small Self

We further comprehensively considered the role of the small self and self-concept clarity when testing the relationship between dispositional awe and self-worth, because they are also two crucial constructs in the self-concept system and, most importantly, are crucially transient states triggered by dispositional awe. Self-concept clarity refers to the extent to which self-beliefs are clearly and confidently defined, internally consistent, and stable [28]. Differences in self-concept clarity show individual differences in clarity, consistency, and confidence in self-beliefs [29]. The term “small self,” which was originally defined as an important consequence of awe, refers to a subjective perception of “self” as being small relative to the vast outside [21,30]. 

According to the small-self hypothesis of awe [20,21], lower self-concept clarity and the smaller self are two pivotal outcomes of dispositional awe. Dispositional awe can significantly alter the self-concept in ways that reflect a shift in attention towards larger entities, a diminishment of the individual self, and an almost metaphorical sense of the smallness of the self [31]. Specifically, people with high dispositional awe tend to perceive something greater than themselves and feel smaller and less significant [21,32], and they are likely to reduce self-awareness and diminish self-concept clarity as they pay more attention to the vast outside than to their ego inside [20,30]. The small-self hypothesis has been adequately supported by abundant empirical evidence. For example, previous studies have demonstrated that diminishment of self-concept and a feeling of smallness are the most important inner mechanisms for conformity, cooperation, and prosocial behavior [20,21,33]. 

Meanwhile, self-concept clarity and the small self are closely associated with self-worth. They all belong to the self-concept system, but they separately describe different aspects of it. Self-worth is an inner core of self-concept, for it refers to the identity of personal values and worth [16,19]. Contingencies of self-worth can facilitate self-regulation and ultimately result in the enhancement of self-esteem [10,34]. Self-concept clarity refers to the extent to which self-beliefs are clearly and confidently defined, internally consistent, and stable [28]. Sufficient theories posit that identity clarity is an important contributor to psychological well-being and higher self-esteem or self-worth [35]. Ultimately, enhancing self-worth requires clarity regarding which goals are most important to people and why [10]. The consistency of self also indicates the prediction of self-concept clarity for a higher sense of self-worth [36,37]. “Small self” is a self-perception of smallness and significance [21], a relatively minor notion for the self-concept that was originally described within awesome situations. A sense of small self implies having less self-value and worth in comparison with the vast world outside [20]. Previous studies also revealed that the small self was positively associated with low self-esteem [31].

In short, comprehensive exploration of self-concept clarity and the small self is a crucial topic when testing the relationship between dispositional awe and self-worth. Despite the limitations of the cross-sectional design, we tend to hypothesize the indirect roles of self-concept clarity and the small self mainly based on the small-self hypothesis of awe and the consistency of self [20,21,36,37]. Based on the above, Hypotheses 2 and 3 were supposed as follows:

**Hypothesis** **2:**
*Greater dispositional awe is positively associated with higher small self and less self-concept clarity.*


**Hypothesis** **3:**
*Greater small self and less self-concept clarity are positively associated with lower self-worth.*


### 1.3. The Hypothesized Model

In sum, the positive effect of dispositional awe has been widely noticed in prosocial behaviors and well-being, but its role in the self-concept system, especially in the inner core of self-worth, has been unclear. The purpose of this study was to test the relationship between dispositional awe and self-worth, comprehensively considering the indirect roles of both self-concept clarity and the small self. This study may be helpful for full comprehension of the positive effect of dispositional awe. Based on the above-mentioned hypotheses, our study theoretically constructs and further examines a latent model, as shown in Figure 1. When focusing on the clarity and perception of self-concept, recent studies have shown that dispositional awe initially results in a diminishment of self-concept and subsequently produces a self-sense of smallness [31]. Thus, we also add a hypothesized path from self-concept clarity to the small self in Figure 1. 

## 2. Research Design and Method

### 2.1. Participants

Using cluster sampling, a total of 2000 undergraduates were recruited from Fuzhou, a southeast coastal city in the P.R.C., and questionnaires were administered to them. Following collection of the questionnaires, all the answers were checked, and the invalid questionnaires, such as those with blanks and/or those answered with the same option in large, were excluded. Finally, 1888 valid questionnaires were retained, and the effective ratio of the questionnaire was 94.4%. Participants’ ages ranged from 18 to 25 years (*M* = 20.7; *SD* = 1.5), of which 618 were males (32.7%) and 1270 were females (67.3%) and 576 were freshmen (30.5%), 456 were sophomores (24.2%), 494 were juniors (26.2%), and 362 were seniors (19.2%). The Academic Committee of ** University from *** Province in the P.R.C reviewed and approved all procedures in this study.

### 2.2. Measures

#### 2.2.1. Dispositional Awe

Dispositional awe was measured by the Dispositional Awe Questionnaire for Chinese Undergraduates (DAQ-CU) developed by Lin and Lian [38]. The DAQ-CU measures the levels of dispositional awe using a multi-factor approach. It consists of 25 items and five factors, including awe of life, nature, relationships, morality, and spirit/religion. All items are rated on a 6-point Likert scale ranging from 1 (does not describe me at all) to 6 (describes me greatly), with higher scores signifying higher levels of dispositional awe. The scale is reliable and valid, Cronbach’s α coefficient for each sub-questionnaire is 0.81~0.88, and Cronbach’s α coefficient for the entire questionnaire is 0.92. In this study, Cronbach’s α coefficient for the total questionnaire was 0.92.

#### 2.2.2. Self-Worth

Self-worth was assessed using Huang and Yang’s general self-worth sub-scale, the self-worth scale for young students [18]. The sub-scale of general self-worth is divided into two dimensions: society-oriented self-worth and person-oriented self-worth, each with five items. The scale employs a 5-point scoring method, ranging from 1 (completely inconsistent) to 5 (completely consistent), with higher scores indicating higher levels of self-worth. The scale has good reliability and validity [18]. Cronbach’s α for social orientation self-worth and personal orientation self-worth were 0.80 and 0.78, respectively, in this study.

#### 2.2.3. Self-Concept Clarity

The Self-Concept Clarity Scale developed by [28], was used to measure self-concept clarity. It is made up of 12 items. All items are scored on a 4-point Likert scale ranging from 1 (very inconsistent) to 4 (very consistent), with higher scores signifying greater self-concept clarity. Cronbach’s α coefficient for this study was 0.83.

#### 2.2.4. Small Self

The Small Self Scale, developed by Preston et al. [39], was used to assess small self. There are 5 items total. All items are rated on a seven-point Likert scale ranging from 1 (completely inconsistent) to 7 (completely consistent), with higher scores indicating higher levels of small self. The scale has good reliability and validity. In this study, Cronbach’s α coefficient was 0.82.

### 2.3. Statistical Analyses

The data were analyzed using SPSS 25.0 for descriptive statistics, correlation analysis, and difference analysis. The Mplus 8.0 software was used to analyze and test the indirect effects of the small self and self-concept clarity, and the bootstrap algorithm was used to estimate the confidence interval of the indirect effect. When formulating the models, all the study variables including dispositional awe, society-oriented self-worth, person-oriented self-worth, small self, and self-concept clarity were considered as latent variables; their measuring indictors were sub-factors or items of the questionnaires. Considered dispositional awe was measured with multiple factors, the latent variable dispositional awe was measured with its five sub-factors that include awe of life, nature, relationships, morality, and spirituality/religion. The other variables were measured with dimensional questionnaires; thus, the latent variables including society-oriented self-worth, person-oriented self-worth, small self, and self-concept clarity were measured with item parcels by item-to-construct balances [40]. All models were estimated by a maximum likelihood estimation procedure. When the evaluated model fit, the chi-square test (*χ2*) and the normed chi-square test (*χ^2^*/*df*) were considered. However, they are significantly influenced by sample sizes [41]. Thus, multiple complementary fit indices, including the root mean square error of approximation (*RMSEA*) and its 90% confidence interval (C.I.), the Tucker-Lewis index (*TLI*), the comparative fit index (*CFI*), and the standardized root mean square residual (*SRMR*) were used to evaluate model fit [42]. The following criteria were used to evaluate fit: CFI and TLI should be equal to or larger than 0.90, and RMSEA and SRMR should be equal to or smaller than 0.08 [41,42].

## 3. Results

### 3.1. Common Method Biases Test

The Harman single factor test was used to examine the common method biases. The results showed that there were nine factors with eigenvalues greater than 1, and the variance contribution rate of the first factor was 19.83%, which is less than 40% [43]. These indicated that the common method biases in this study were not severe.

### 3.2. Descriptive Statistics and Correlation Analysis

Table 1 displays the descriptive statistics and reliability and validity test results of the research variables. Dispositional awe was positively correlated with both society-oriented and individual-oriented self-worth (*p_s_* < 0.001). The first hypothesis was preliminarily supported. Dispositional awe was negatively associated with self-concept clarity (*p* < 0.001) and positively correlated with small self (*p* < 0.001). The second hypothesis received preliminary support. Self-concept clarity was positively correlated with social- and person-oriented self-worth (*p_s_* < 0.001), whereas small self was negatively correlated with both (*p_s_* < 0.001). The third hypothesis was preliminarily supported.

### 3.3. Direct and Indirect Effects Test

A structural equation modeling approach was used to test the direct and indirect effects of dispositional awe on self-worth. For each latent variable, the following observation indicators and project packaging strategies were used: The dispositional awe is a multidimensional result, with the total score of each dimension regarded as its observation indicator via dimensional packaging [44,45]. One-dimensional structures include the small self, society-oriented self-worth, and person-oriented self-worth. The number of their measurement items is small, so the items are considered observation indicators [44,45]; self-concept clarity is one-dimensional, and there are many measurement items (12 items), so the items were parceled as three packages by the method of item balance [44,45].

Confirmatory factor analysis (CFA) was used to test the measurement model. Considered a larger sample in our study, the *χ^2^*/*df* value was larger than 5, and the value of TLI was smaller than 0.90, but the other fitting indexes all met the criteria for good fitting (*χ^2^*/*df* = 7.17, RMSEA = 0.06, SRMR = 0.09, CFI = 0.90, TLI = 0.88), indicating that the set latent variables were well measured by the measurement indexes.

Firstly, without considering self-concept clarity and the small self, a direct path model was constructed to test the direct effect (total effect) of dispositional awe on person-oriented and society-oriented self-worth. The direct path model fit the data well (*χ^2^*/*df* = 11.65, RMSEA = 0.07, SRMR = 0.07, CFI = 0.93, TLI = 0.91), and dispositional awe was positively associated with person-oriented and society-oriented self-worth (*β* = 0.24, 0.45, *SE* = 0.03, 0.03, *p_s_* < 0.001).

Secondly, the indirect roles of self-concept clarity and small self in the relationship between dispositional awe and person-oriented and society-oriented self-worth were further tested. The results demonstrated that the full model, including the indirect variables of self-concept clarity and small self, adequately met the standard of good fitting (*χ^2^*/*df* = 10.32, *RMSEA* = 0.07, *SRMR* = 0.07, *CFI* = 0.90, *TLI* = 0.88). See Figure 2 and Table 2 for the bootstrap confidence interval estimation of the standardized path load and indirect effect of the whole model.

Notice: Gender and age were controlled. Dispositional awe was measured with its five sub-factors including awe of life, nature, relationships, morality, and spirit/religion. Society-oriented self-worth, person-oriented self-worth, small self, and self-concept clarity were measured with item parcels by item-to-construct balances. 

When the indirect roles of the small self and self-concept clarity were taken into account, the direct predictive effect of dispositional awe on person-oriented self-worth increased but not significantly (0.50 vs. 0.45, *t* = 1.67, *p* = 0.343). As illustrated in Figure 1, dispositional awe was positively correlated with the small self (*β* = 0.21, *p* < 0.001), whereas the small self was negatively associated with person-oriented self-worth (*β* = −0.33, *p* < 0.001). The indirect effect of small self was significantly negative (Es = −0.07, 95% *CI* = [−0.11, −0.04]), which was contrary to the direct effect of dispositional awe; therefore, it should be explained as a suppressing effect [46]. The absolute value ratio of the suppressing effect to the direct effect was 11.7%. 

Similarly, dispositional awe was negatively correlated with self-concept clarity (*β* = −0.08, *p* = 0.008), self-concept clarity was negatively correlated with small self (*β* = −0.60, *p* < 0.001), and small self was also negatively correlated with person-oriented self-worth. The chain indirect effect of both self-concept clarity and small self was significant (Es = −0.02, *95%CI* = [−0.03, −0.01]), which was also in opposition to the direct effect of dispositional awe; therefore, it should also be explained as a suppressing effect [46]. The absolute value ratio of the suppressing effect to the direct effect was 3.33%.

After controlling for the indirect roles of the small self and self-concept clarity, the direct predictive effect of dispositional awe on society-oriented self-worth increased significantly (0.36 vs. 0.24, *t* = 4.00, *p* < 0.001). Small self and self-concept clarity had significantly negative indirect effects (Es = −0.09, −0.01, *95%CI* = [−0.13, −0.06], [−0.02 −0.00]), which were the inverse of dispositional awe’s direct effect and thus should be accounted for as a suppressing effect [46]. The absolute value ratio of the suppressing effects to the direct effect was 18.8% to 2.1%, respectively. The indirect chain effect of self-concept clarity and small self was significantly negative (Es = −0.02, *95%CI* = [−0.04 −0.01]) and should also be interpreted as a suppressing effect, accounting for 4.2% of the absolute value of the direct effect.

## 4. Discussion

The main goal of our study was to explore the relationship between dispositional awe and self-worth in Chinese undergraduates, which comprehensively considered two variables related to self-concept, namely, self-concept clarity and the small self. Overall, our study revealed a positive correlation between dispositional awe and self-worth. However, two significant transcendental states of dispositional awe—small self and diminishment of self-concept—suppressed this relationship. This study was the first one to explore the relationship between dispositional awe and multiple factors of self-concept, which may make it easier to fully comprehend how dispositional awe affects the self-concept system.

### 4.1. The Effect of Dispostional Awe on Self-Worth Is Positive

The first question considered in our study was whether the association between dispositional awe and self-worth is positive or negative. The answer is positive. The results showed that dispositional awe was positively correlated with both society-oriented and person-oriented self-worth among Chinese undergraduates (*r* = 0.12, 0.27). The positive correlations of dispositional awe with person-oriented and society-oriented self-worth were 0.24 and 0.45, respectively. This study is the first to demonstrate and support the beneficial impact of dispositional awe on one’s sense of worth. This study tends to support the self-transcendent trait of dispositional awe rather than the trait of diminishment of self [21,23]. As demonstrated by well-being and prosocial behaviors [27,47], this study supports a positive effect of dispositional awe on self-worth, encompassing society-oriented and person-oriented aspects. The explanation for the results is the self-transcendent model of awe [23]. Dispositional awe, a self-transcendent affect, is particularly effective at creating social resources due to its ability to bond individuals together and thus ultimately help people realize their self-worth. These findings not only reaffirm the self-transcendent trait of dispositional awe [25] but also indicate that dispositional awe is advantageous for the inner realization of self-worth and ultimately raises self-esteem. 

### 4.2. The Indirect Effect of Small Self and Self-Concept Clarity Are Suppressing

The second focus in this study was to further consider the roles of both self-concept clarity and the small self in the relationship between dispositional awe and self-worth. The results found their suppressing effects during this association, for the indirect effects were contrary to the positive direct effect (suppressing effects = −0.01, −0.02; −0.09, −0.01, −0.07, −0.02; see Figure 2 and Table 2). These findings contribute to our understanding of the function of dispositional awe in the self-concept system. 

First, this study found different effects of dispositional awe on self-worth, self-concept clarity, and small self. As shown in the model (Figure 2), the effects of dispositional awe on self-worth, including society-oriented and person-oriented aspects, were positive, but on self-concept clarity and small self, they were negative. These results are similar to previous studies that considered dispositional awe from positive and negative perspectives [26,48]. The complexity of dispositional awe itself is largely responsible for the various roles. Apart from vastness and accommodation, the main ingredients of dispositional awe also vary in additional appraisals and/or flavoring elements, such as threat, beauty, ability, virtue, and the supernatural [1]. There has always been some evidence that dispositional awe consists of a small amount of fear and threat [48,49]. The positive and minor negative components of dispositional awe show different impacts on the self-concept system. 

Furthermore, the suppressing effects of self-concept clarity and small self can be explained by the small self hypothesis of dispositional awe and the self-consistency theory. As adequately evidenced by the small-self hypothesis, diminishment of self-concept and perception of smallness are pivotal results of dispositional awe, and self-concept clarity and the small self are contrary to higher self-worth, which is mainly held up by the consistency of self. Last, but most importantly, the total positive effect of dispositional awe on self-worth manifests as positive significance in self-concept development. Though dispositional awe provisionally diminishes self-identity and reduces the perception of self, it can ultimately urge people to overcome their existing smallness, seek alignment with the collective or the outside world, focus more on collective input and the authentic self, and thus achieve greater self-worth [27]. 

### 4.3. Implications and Limitations in This Study

In sum, this study supports a positive association between dispositional awe and self-worth, although this relationship is inhibited by self-concept clarity and the small self. This study helps us fully comprehend the trait of dispositional awe and its significance for self-concept development. First, while dispositional awe temporarily reduces self-concept clarity and perception of self, it eventually leads to higher self-worth. Constitutionally, dispositional awe is positive and self-transcendent. Furthermore, this research suggests that cultivating dispositional awe benefits people’s sense of self-worth and thus fosters the growth of their self-concept. During this process, diminishing self and the smallness of self-concept should also be noticed. It is significant for educators and parents to foster adolescents’ dispositional awe, which benefits their self-concept and/or self-awareness development. 

This study has several limitations. First, this study only sampled Chinese undergraduates; further studies should include a diverse range of ages and cultures in the samples to validate the findings. Secondly, this study was of a cross-sectional design, so it cannot support a causal association. Further research should clarify which is the cause and which is the result of the relationship between dispositional awe and self-worth and test the indirect roles with longitudinal studies and/or an experimental design. Moreover, this study only considers self-worth, which is essential for self-concept, but the self and/or self-concept systems are complex and contain many aspects from varied perspectives, for example, from the self-construal perspective. Self-construal involves individual, relational, and collective levels [50]. More studies should be done to investigate the role of dispositional awe in other aspects of self-concept.

## 5. Conclusions


Dispositional awe was positively correlated with society-oriented self-worth and person-oriented self-worth.Dispositional awe was positively correlated with the small self and self-concept clarity, but the small self and self-concept clarity were negatively correlated with society-oriented self-worth and person-oriented self worth. The indirect effects of small self and self-concept clarity and the chain indirect effect were explained as suppressive effects.


This study suggested that dispositional awe could help people better understand themselves and enhance their sense of self-worth.

## Figures and Tables

**Figure 1 ijerph-20-06296-f001:**
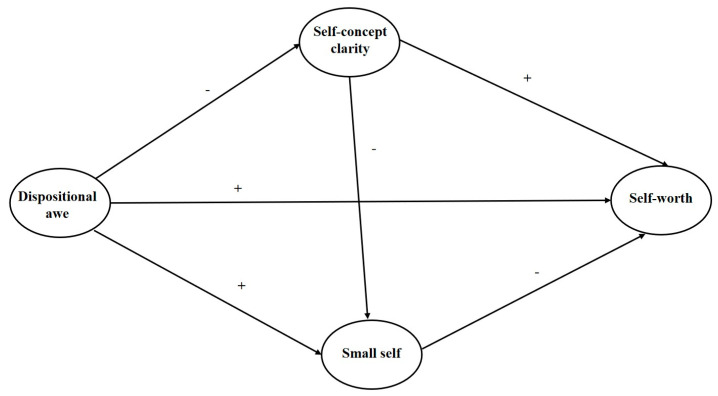
The hypothesized model.

**Figure 2 ijerph-20-06296-f002:**
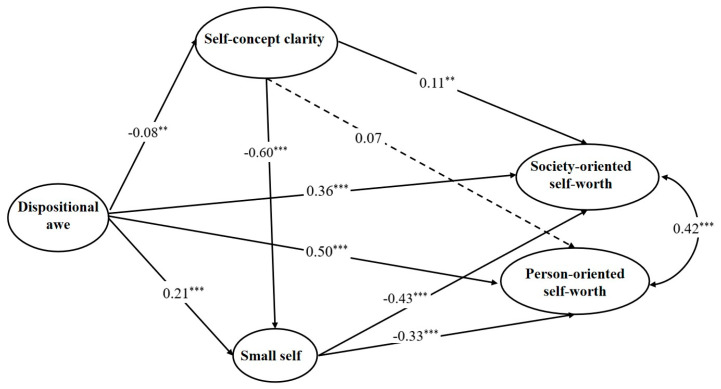
The indirect effect model of small self and self-concept clarity. ** *p* < 0.01; *** *p* < 0.001.

**Table 1 ijerph-20-06296-t001:** Pearson correlation of research variables (*n* = 1888).

Variables	1	2	3	4	5
1 Dispositional awe	**0.72**				
2 Self-concept clarity	−0.18 ***	**0.58**			
3 Small self	0.33 ***	−0.47 ***	**0.69**		
4 Society-oriented self-worth	0.12 ***	0.31 ***	−0.29 ***	**0.70**	
5 Person-oriented self-worth	0.27 ***	0.25 ***	−0.16 ***	0.54 ***	**0.67**
*M*	115.94	32.96	23.03	17.49	18.03
*SD*	15.57	6.86	5.98	3.41	3.16
CR	0.96	0.84	0.80	0.82	0.79
AVE	0.52	0.34	0.47	0.50	0.45

Note: CR = composite reliability, AVE = average variance extracted; *** *p* < 0.001. The values on the diagonal were squares of the AVE.

**Table 2 ijerph-20-06296-t002:** Bootstrap estimation of the indirect effects of self-concept clarity and small self.

Path	Standardization Estimated Value	*SE*	Bootstrap 95% Lower	Bootstrap 95% Higher	Accounting for the Total Effect (%)
Dispositional awe → society-oriented self-worth	0.36	0.03	0.30	0.40	75.0
Dispositional awe → self-concept clarity → society-oriented self-worth	−0.01	0.01	−0.02	−0.00	2.1
Dispositional awe → small self → society-oriented self-worth	−0.09	0.02	−0.13	−0.06	18.8
Dispositional awe → self-concept clarity → small self → society-oriented self-worth	−0.02	0.01	−0.04	−0.01	4.2
**Total effect**	0.24	0.03	0.17	0.29	-
Dispositional awe → person-oriented self-worth	0.50	0.03	0.44	0.55	83.3
Dispositional awe → self-concept clarity → person-oriented self-worth	−0.01	0.00	−0.02	0.00	1.7
Dispositional awe → small self → person-oriented self-worth	−0.07	0.02	−0.11	−0.04	11.7
Dispositional awe → self-concept clarity → small self → person-oriented self-worth	−0.02	0.01	−0.03	−0.01	3.3
**Total effect**	0.42	0.03	0.35	0.46	-

## Data Availability

The datasets generated during and/or analyzed during the current study are available from the corresponding author on reasonable request.

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
