# Peer review of "Dispositional Awe and Self-Worth in Chinese Undergraduates: The Suppressing Effects of Self-Concept Clarity and Small Self"

_ijerph, 2023, doi:10.3390/ijerph20136296_

Round 1

Reviewer 1 Report

Authors need to:

Add research design since the y have used methods from different sources

Revisit discussion and  add conclusions

References are different from the   MDPI requirements

Author Response

 Thanks very much for giving us their constructive suggestions, which will help us both with our English and to improve the quality of the work. 

Here, we have submitted a new version of our manuscript that has been modified according to your suggestions. Efforts have been made to correct the mistakes and improve the English in the manuscript. We have marked all changes in Blue in the revised manuscript.

The following is a point-by-point response to the reviewer’ comments.

Authors need to:

Add research design since they have used methods from different sources

Revisit discussion and  add conclusions

Response: Thanks very much for the comment. we have added the conclusion section in our revised version.

And we also revised the discussion section according to the other reviewers.

Research design has been added.

References are different from the   MDPI requirements

Responses: all the references have been revised according to the MDPI requirements.

Reviewer 2 Report

Congratulation for the aauthors for their valuable work.I like how you structured the hypotheses, the study design. A short conclusion would be beneficial.

- Three hypotheses are presented by the authors, but the purpose of the study is unclear
- What is the novelty of this research?
- What is the clinical relevance of this study?
- The conclusions are missing.

Thank you.

Author Response

Thanks very much for the reviewer's suggestion!

all the revised have been colored with blue. 

Congratulation for the aauthors for their valuable work.I like how you structured the hypotheses, the study design. A short conclusion would be beneficial.

Response: Thanks very much for the reviewer’s positive comment. According to the suggestion, a short conclusion have added in our revised version.
- Three hypotheses are presented by the authors, but the purpose of the study is unclear

Response: Thanks very much for the reviewer’s positive comment. The purpose of this study was to tests the relationship between dispositional awe and self-worth, comprehensively considering the indirect roles of both self-concept clarity and the small self.

We has added it in the last paragraph in the introduction.
- What is the novelty of this research?

Response: Thanks very much for the reviewer’s positive comment. the novelty of this study has added in the first paragraph in the discussion.
- What is the clinical relevance of this study?

Response: Thanks very much for the reviewer’s comment. the clinical relevance of this study has added in the last paragraph in the discussion.
- The conclusions are missing.
Response: the conclusion section has added.

Reviewer 3 Report

While I find this article interesting, it is necessary for the authors to adhere rigorously to these major revisions in order for this article to be published:

Abstract:

-       The abstract should be divided into different parts as indicated in the instructions for authors in this journal.

Keywords:

-       All keywords are repeated in the title of the article. I strongly recommend to remove them and look for other key words that help to search this article better to give it a better transfer.

Introduction:

-       The introduction is excellent, but authors are advised to formulate a sentence with the hypothesis connecting it to the background and justifications instead of writing it "aside" without establishing a connection with the text. An example to rephrase it would be "Based on the above data, the main hypothesis is...".

Methodology..

-       It is necessary to include the criteria for inclusion and exclusion of study participants, as well as to further specify the statistical analyses, as the section is insufficient.

Discussion:

-       When talking about correlations, it is convenient to include the exact data as well as the degree of relationship and the quantity. It needs to be included.

-       The overall discussion needs to be improved considerably for this article to be accepted as it is merely a summary of the results. You should find similarities and differences with results obtained in other similar studies and explain the reasons for these discrepancies or similarities as well as justify your own results. Practical implications of this study as well as future lines of research should also be included separately.

Conclusion

-        The conclusion section is missing, both in terms of format and as a summary of the most important findings of this study.

Author Response

see in the attached. 

Reviewer 4 Report

This study explores the relationship between awe and self-worth among Chinese university students, as well as the role of self-concept clarity and self-enhancement in this association. The overall writing of the paper is relatively standard, but the following revisions are needed before making a decision on acceptance for publication.

  1. Figure 1 does not display the controlled variables for the theoretically constructed and further tested latent model. The authors should explain how they addressed the issue of controlling variables in this study.
  2. Figure 1 lacks a title.
  3. "2. Method" starts directly with "2.2. Participants," missing 2.1. Is this a typographical error or did the authors omit a portion of the content?
  4. The authors mention that "invalid questionnaires, such as those with blanks and regular answers, were excluded." Why would regular answers be excluded? Shouldn't normal responses be retained?
  5. The authors state, "The scale is reliable and valid, Cronbach's α coefficient for each sub-questionnaire is 0.81~0.88, and Cronbach's α coefficient for the entire questionnaire is 0.92. In this study, Cronbach's α coefficient for the total questionnaire was 0.92." Why are the Cronbach's α coefficients for the sub-questionnaires lower than the Cronbach's α coefficient for the total questionnaire? This discrepancy raises concerns about the reliability of the data.
  6. The meaning of the values on the diagonal of Table 1 is unclear. Typically, the diagonal reports the correlation coefficients, so the values on the diagonal should be 1. If the values on the diagonal represent means, it should be explicitly specified below the table.
  7. The fit of the structural equation model constructed by the authors is not satisfactory. The χ2/df should be less than 3, and the TLI should be greater than 0.9. If these conditions are not met, the authors should clearly state the reasons. Additionally, I suggest that the authors report the critical values for different fit indices together.

8.      In addition, it is suggested to refer to some empirical studies related to the topic. For instance,(1)Does attending elite colleges matter in the relationship between self-esteem and general self-efficacy of students in China?. Heliyon. 8. e09723. 10.1016/j.heliyon.2022.e09723.(2)Longitudinal Relationship Between Self-Esteem and Academic Self-Efficacy Among College Students in China: Evidence From a Cross-Lagged Model. Frontiers in psychology, 13, 877343. https://doi.org/10.3389/fpsyg.2022.877343

 Moderate editing of English language required

Author Response

see in the attached. 

Round 2

Reviewer 3 Report

The authors have resolved the proposed suggestions

Reviewer 4 Report

The author has made changes in response to comments and I have no further comments.